# The Impact of Digital Financial Inclusion on the Level of Agricultural Output

**Sheng Xu [1,2,\*] and Jingwen Wang [1]**

1 School of Economics, Ocean University of China, Qingdao 266100, China
2 Institute of Marine Development, Ocean University of China, Qingdao 266100, China
\* Correspondence: xusheng@ouc.edu.cn

**Abstract:** The development of digital inclusive finance has alleviated the problem that traditional finance cannot fully cover rural areas, provided convenient services for Chinese farmers, and solved the problem of "difficult and expensive loans" in agricultural development. This paper used the panel data of Beijing University's Digital Inclusive Finance Index and 31 provinces and cities in China from 2011 to 2020, and adopted the double-fixed-effect and panel threshold model to study the impact of the development of digital inclusive finance on the level of agricultural output and its internal mechanism. The study found that digital inclusive finance can significantly improve the level of agricultural output, and there is a double threshold for the impact of digital inclusive finance on the level of agricultural output. The heterogeneity analysis showed that the coverage and depth of digital inclusive finance can significantly improve the level of agricultural output, and the depth of use plays a greater role. Digital inclusive finance has significantly improved the level of agricultural output in the midwest regions and major agricultural provinces, but its impact on the eastern regions and non-agricultural provinces is not significant. Finally, the mechanism analysis found that digital inclusive finance can improve the level of agricultural output by promoting the level of agricultural mechanization and improving farmers' willingness to participate in insurance. Therefore, we should continue to promote the development of digital inclusive finance according to local conditions.

**Keywords:** China; digital inclusive finance; agricultural output level; threshold effect; heterogeneity analysis





## 1. Introduction

Stabilizing the supply of agricultural products is the cornerstone of rural economic development and an important guarantee for the implementation of the rural revitalization strategy. China is a traditionally agricultural country. Since the 21st century, "Central Document No. 1" has focused on the "Three Rural Issues" for 19 consecutive years. China's food output increased from 490 million tons in 2000 to 682.85 million tons in 2021, and the gross agricultural output value also increased from USD 1675.88 billion to USD 12,143.40 billion, according to the National Bureau of Statistics of the People's Republic of China. China is also a large consumer of grain. According to the General Administration of Customs, China imported 164.539 million tons of grain in 2021, accounting for 24% of China's output, which reached a record high, indicating that China's dependence on foreign food is also increasing. In addition, the transfer of young laborers, the aging of the population, the reduction in cultivated land area, and the foreign investors' control of seed sources are constraining the development of agriculture [1–4]. It is of great practical significance to increase food output, stabilize the supply of agricultural products to consolidate the achievements of poverty alleviation, and realize the strategy of rural revitalization.

With the rise of a new round of information and technology revolutions, digital inclusive finance, characterized by the use of emerging technologies, has become a new driving force for China's inclusive economic growth. In 2016, the Hangzhou G20 Summit officially

put forward the concept of digital inclusive finance, referring to "all actions to promote inclusive finance through the use of digital financial services". In 2021, the "Document No. 1"—"Central Committee of the Communist Party of China and the State Council's Opinions on Promoting Rural Revitalization and Accelerating Agricultural and Rural Modernization" put forward the "Development of Rural Digital Inclusive Finance" for the first time. As a new model and industry formed by the integration of traditional finance with the new generation of information and communication technology and internet-related technology [5], digital inclusive finance has created a new plan for the development of the rural economy and the service of rural revitalization. The birth of digital inclusive finance overcomes the difficulty of traditional inclusive finance in reconciling social and commercial benefits, helps to alleviate the spatial and geographical constraints, reduces the service threshold of finance in rural areas, reduces the high cost of financial services in rural areas, adjusts the allocation of urban and rural financial resources effectively, narrows the gap of financial services between urban and rural areas, and brings an opportunity to promote the development of agriculture and improve the level of agricultural output. According to the latest internet development report—"The 50th statistical report on China's Internet Development"—as of June 2022, the internet penetration rate in China's rural areas has been close to 60%, with 1.051 billion internet users, and the number of rural internet users has reached 293 million, accounting for 27.9% of the total number of netizens. Internet infrastructure construction in rural areas is being comprehensively strengthened, with "counties connected to 5G and villages connected to broadband"; digital technologies such as big data, artificial intelligence, and cloud computing have deepened the integration with planting, animal husbandry, and fisheries; rural e-commerce has developed rapidly, and the online retail of agricultural products has increased by 11.2%. The popularization of the internet in rural areas has laid a solid foundation for digital inclusive finance to help rural revitalization and serve farmers' production.

With the rapid promotion of digital inclusive finance in rural areas, academia has begun to pay attention to the impact of digital inclusive finance on rural economic development. Existing research shows that digital inclusive finance has a driving effect on the development of the rural economy [6]. From the macro level, the development of digital inclusive finance is conducive to optimizing the agricultural industrial structure [7,8], improving the level of agricultural mechanization [9,10], and promoting the integration of three rural industries [11,12], improving agricultural ecological efficiency [13], and promoting high-quality agricultural development [14,15]. From the micro level, digital inclusive finance can improve farmers' credit access, promote farmers' entrepreneurship [16–18], and narrow the income gap and consumption gap between urban and rural residents [19–22]. However, there are still relatively few studies on the impact of digital inclusive finance on agricultural output and its mechanism. Based on the digital inclusive finance index compiled by Peking University and the China Statistical Yearbook database, this paper empirically studies the impact of digital inclusive finance on agricultural output and its underlying mechanism using the double-fixed-effect and panel threshold model.

The possible marginal contributions of this paper are: First, based on the panel data of 31 provinces from 2011 to 2020, the panel threshold model confirmed that the impact of digital inclusive finance on agricultural output level shows an increase in marginal utility, which provides evidence for digital inclusive finance to promote agricultural production. Second, the paper analyses the structural heterogeneity of the impact of digital inclusive finance on agricultural output levels at the national level, and further divides the total sample into eastern and midwest regions, agricultural provinces and industrial provinces to demonstrate the regional heterogeneity and scale heterogeneity of the impact of digital inclusive finance on agricultural output levels, and to provide support for the formulation of digital inclusive finance differentiated development policies. Finally, this paper argues the specific path of digital inclusive finance's effect on agricultural output level from agricultural mechanization and farmers' participation in insurance, which expands the

research on the paths of digital inclusive finance's effect on agricultural output and makes up for the shortcomings of the existing literature.

The rest of this article is arranged as follows. Section 2 is the theoretical analysis and research hypothesis. Section 3 introduces the research model and describes the data. Section 4 analyzes the empirical results. Section 5 introduces the research conclusions, policy recommendations and limitations of the research.

## 2. Theoretical Analysis and Research Hypothesis

Relying on the advantages of digital technology, digital inclusive finance alleviates the spatial and geographical constraints away from the commercial outlets, lowers the threshold of rural financial services, and brings opportunities to promote agricultural development and improve agricultural output.

### 2.1. Direct Impact of Digital Inclusive Finance on the Agricultural Output Level

There is no doubt that the development of agriculture cannot be separated from financial support. Some studies found that under the restriction of geographical distance and the exclusion of traditional finance, credit constraints have become the main reason for hindering agricultural development [23]. Digital inclusive finance is an emerging form of financial business that combines digital technology with financial activities. It integrates advanced technologies such as big data, artificial intelligence, the Internet of Things, and blockchain. For financial institutions, digital inclusive finance can, on the one hand, free traditional financial institutions from relying on physical outlets, reduce the manpower and construction cost required to set up outlets, shorten the supply time of financial services, and improve the efficiency of financial services [24]. On the other hand, digital inclusive finance can alleviate the risk caused by information asymmetry, realize the use of big data to understand the credit status of farmers before lending, decide whether to issue loans and reduce adverse selection. After lending, we can use the internet and big data to understand the destination of funds and assess the credit rating of customers in real time, as well as to mitigate the losses caused by moral hazards to financial institutions [25]. For farmers, digital inclusive finance lowers the threshold of financial services, enables farmers to obtain financial services fairly and reasonably, alleviates capital constraints and high loan costs in the process of large-scale production, and expands the source of funds for farmers' large-scale operations so that it is easier to win the favor of farmers [26,27]. As pointed out by Abdulai [28] and Kikulwe [29], the use of mobile payment will increase the possibility of farmers purchasing fertilizers and pesticides, which will positively impact agricultural production. Peprah [30] also found that the crop yield of farmers using digital payment will increase by 33.8% to 56% through sampling surveys in three rural areas of Ghana.

In addition, China has a vast territory, and the economic development of regions is different; therefore, the basis for the development of digital inclusive finance is also different. The overall development of digital inclusive finance is nonlinear, and there is a nonlinear spillover effect on agriculture [31,32]. At the initial stage of the development of digital inclusive finance, the infrastructure in remote areas was relatively backward, and farmers generally lacked the understanding of digital inclusive finance; thus, the impact of digital inclusive finance on agriculture was relatively small. With the improvement of the infrastructure and the implementation of relevant policies, the development of digital inclusive finance has been supported by technology and policy, and its effect on agricultural production has been further enhanced. Based on the above analysis, this paper puts forward hypothesis 1:

**Hypothesis 1 (H1).** *Digital inclusive finance can contribute to promoting the improvement of agricultural output, and there is a non-simple linear relationship between them.*

### 2.2. Influence Mechanism of Digital Inclusive Finance on the Agricultural Output Level

Digital inclusive finance promotes higher levels of agricultural output by improving the level of mechanization in agricultural production and increasing farmers' willingness to participate in insurance.

#### 2.2.1. Improving the Level of Agricultural Mechanization

The mechanization of agricultural production can not only improve the efficiency of agricultural production but also increase food production [33]. When plowing, the use of machinery can be more efficient in completing the work. Especially for farmers who have large farms, mechanization not only reduces labor costs but also helps to promote standardized agricultural production. When sowing and fertilizing, the use of agricultural machinery will make the distribution of crops and fertilizers more even, improve the utilization rate of fertilizer, and achieve the purpose of increasing production. When harvesting, mechanized production can achieve the effect of "resisting disasters and seizing the farming season" and reduce losses caused by natural disasters. At the same time, agricultural mechanization also solves the problem of labor shortage in peak season and promotes agricultural production by alleviating the human bottleneck in agricultural production [34–36]. Aryal [37] pointed out that agricultural mechanization is a boon for small farmers in developing countries, especially in areas with less cultivated land per capita. The development of agricultural mechanization has alleviated the disadvantages of the large population, small land, and low food production. However, the lagging development of rural finance has made rural financial exclusion widespread. A high level of financial exclusion will inhibit the promotion of agricultural science and technology in the agricultural economy [38]. Digital inclusive finance can effectively alleviate financial exclusion, with financing and income-increase effects [39]. On the one hand, digital inclusive finance can alleviate the potential risks in the operation of financial institutions, increase the possibility of loans to farmers, and expand the source of funds for farmers to purchase agricultural machinery. On the other hand, the development of digital inclusive finance can increase farmers' income, improve farmers' investment in fixed assets in the agricultural production process, and promote agricultural mechanization.

**Hypothesis 2 (H2).** *Digital inclusive finance increases the level of agricultural output by promoting agricultural mechanization.*

#### 2.2.2. Increasing Farmers' Willingness to Participate in Insurance

Agricultural insurance can mitigate the impact of natural disasters on agricultural production, compensate for losses caused by irresistible factors, and transfer risks in agricultural production. It can also replace some mortgages [40], reduce the risk of loans, expand the scale of rural credit, enhance farmers' confidence in production [41], motivate farmers to expand cultivation, increase crop yields, and promote the development of the rural economy [42–44]. However, the existence of moral hazards will reduce the role of agricultural insurance. The rise of digital inclusive finance can reduce the information asymmetry between insured farmers and insurance companies with the help of the internet, big data, and other technologies. For insurance companies, digital inclusive finance, based on digital technologies, can accurately portray insured people and understand customer needs [45]. At the same time, it can quickly target the disaster area in combination with satellite images, reduce assessment costs and improve the efficiency of claims' settlement [46]. For farmers, digital inclusive finance can popularize knowledge of agricultural insurance, improve farmers' financial literacy [47], and enhance their willingness to participate in insurance. At the same time, it can enable farmers to adopt online insurance to break through temporal and regional restrictions and improve the convenience of insurance participation.

**Hypothesis 3 (H3).** *Digital inclusive finance improves agricultural output by increasing the participation rate of farmers.*

## 3. Research Methods and Data Sources

### 3.1. Methods

In order to test the impact of digital inclusive finance on the level of agricultural output and test Hypothesis 1, the following model is constructed in this paper:

$$Lnoutp_{it} = \alpha_0 + \alpha_1 Lndf_{it} + \alpha_2 LnX_{it} + \mu_i + \lambda_t + \varepsilon_{it} \tag{1}$$

where $i$ represents the province, $t$ represents the year, $outp_{it}$ represents the level of agricultural output, $df_{it}$ represents the digital inclusive financial index, $X_{it}$ are control variables, including crop-planting area, irrigation area, fertilizer use, natural disaster rate, human capital, and financial support for agriculture, $\mu_i$ is the individual fixed effect, $\lambda_t$ is the time fixed effect, $\varepsilon_{it}$ is a random error term.

Digital inclusive finance inherently has non-equilibrium in its development and may have a non-linear impact on the level of agricultural output. This paper constructs a threshold model with digital inclusive finance as the threshold variable:

$$Lnoutp_{it} = \beta_0 + \beta_1 Lndf_{it} \times I(Lndf_{it} \leq d) + \beta_2 Lndf_{it} \times I(Lndf_{it} > d) + \beta_3 LnX_{it} + \mu_i + \lambda_t + \varepsilon_{it} \tag{2}$$

where $d$ is the threshold value of digital inclusive finance, and $I(\cdot)$ is an indicator function. The value depends on whether the level of digital inclusive finance development meets the threshold conditions in parentheses. If it does, it is assigned a value of 1; otherwise, it is 0.

### 3.2. Variables

Dependent Variables: Agricultural output level ($outp_{it}$). In this paper, the gross output value of agriculture, forestry, animal husbandry, and fishery production is chosen to represent the level of agricultural output. Some scholars use the gross output value per capita of agriculture, forestry, animal husbandry, and fishery to represent the level of agricultural output, but Zhang et al. [48] pointed out that the migration of migrant workers to urban areas affects the accuracy of the measurement of agricultural workers, and the application of modern technology in the agricultural production process replaces the labor force. The measurement error is larger when using the per capita output value.

Independent Variables: The development level of digital inclusive finance ($df_{it}$). This paper adopts the digital inclusive finance index (2011–2020), which has been measured by the research team of the Digital Finance Research Center of Peking University and Ant Group Research Institute since 2016. The index of digital inclusive financial includes three dimensions and 33 specific indicators. For further study, this paper also selects the breadth of coverage ($df\_bre_{it}$) and depth of use ($df\_dep_{it}$) of the digital inclusive financial index to study its effects on the level of agricultural output. Among them, the breadth of coverage ($df\_bre_{it}$) is measured by the proportion of bank cards tied to third-party accounts, such as WeChat and Alipay, and their coverage. The more bank cards bound to third-party accounts, the wider the coverage. The depth of usage ($df\_dep_{it}$) measures the actual use of digital inclusive financial services in daily life.

Control variables: To control the influence of relevant factors on the level of agricultural output as much as possible, the following control variables are set in this paper concerning the relevant literature. Crop-sown area ($sarea_{it}$): Unlike arable land, which may be abandoned or semi-abandoned, crop-sown area truly reflects the actual area of crops planted and should have a positive correlation with the level of agricultural output. Irrigation area ($irri_{it}$): Proper irrigation of crops will help crops grow and increase crop yields. Fertilizer usage ($fer_{it}$): In the process of agricultural production, chemical fertilizers can eliminate the damage of pests to crops and improve the level of agricultural output, which is expressed in this paper using the discounted fertilizer application amount. The natural disaster rate level ($dis_{it}$): The occurrence of natural disasters will affect the growth of crops, resulting in crop yield reduction. In this paper, the ratio of damage area (the sown area of crops with more than 10% reduction in production due to disasters) and disaster-caused area (the sown area of crops with more than 30% re-duction in production

due to disasters) to the total planting area of crops is weighted by 0.1 and 0.3 to characterize the level of natural disaster rate. Financial support to agriculture ($fis_{it}$). In this paper, the expenditure on agriculture, forestry, and water in the public expenditure of each province is selected to represent the government's support for agricultural production. Human capital ($edu_{it}$): Farmers with a high level of education tend to be more receptive and operate high-tech products, which, in turn, increases labor productivity and output level. Therefore, this paper uses the weighted years of education in rural areas to measure human capital.

*3.3. Data*

This paper selects panel data of 31 provinces, municipalities, and autonomous regions (excluding Hong Kong, Macao, and Taiwan) in China from 2011 to 2020, with a total of 310 samples to study and analyze the impact of digital inclusive finance on the level of agricultural output. The data include two parts: agricultural production-related data, and digital inclusive financial data. Among them, agricultural production-related statistics are from China Statistical Yearbook (2011–2020), China Rural Statistical Yearbook (2011–2020), China Population and Employment Statistical Yearbook (2011–2020) and China Agricultural Machinery Industry Yearbook (2011–2020). Digital inclusive finance-related indicators are from the Digital Finance Research Center of Peking University. This paper mainly selects the digital inclusive finance index, coverage breadth and usage depth. To eliminate the influence of price factors, this paper uses the GDP deflator to deflate price-related variables in 2010 as the base period since logarithms will not only not change the nature and related relationships of the data, but also prevent the impact of extreme outliers and alleviate heterogeneity. Therefore, referring to the relevant literature [11,49,50], some variables (except *Dis*) are logarithmically processed to keep the data smooth. The descriptive statistics of each variable are shown in Table 1.

**Table 1.** Descriptive statistics of key variables.

|  | Variable Name | Meaning | Observations | Mean | Min | Max |
|---|---|---|---|---|---|---|
| Dependent Variables | *Lnoutp* | Total output value of agriculture, forestry, animal husbandry and fishery production (CNY 100 million ) | 310 | 7.5231 | 4.6487 | 9.1522 |
| Independent Variables | *Lndf* | Digital inclusive financial index | 310 | 5.2116 | 2.7862 | 6.0683 |
|  | *Lndf_bre* | Coverage of digital inclusive finance | 310 | 5.0596 | 0.6729 | 5.9839 |
|  | *Lndf_dep* | Use depth of digital inclusive finance | 310 | 5.1948 | 1.9110 | 6.1917 |
| Control variables | *Lnsarea* | Crop-sown area (thousand hectares) | 310 | 8.1086 | 4.4841 | 9.6098 |
|  | *Lnfer* | Amount converted from fertilizer application (10,000 tons) | 310 | 4.7225 | 1.4816 | 6.5738 |
|  | *Lnirri* | Irrigation area (thousand hectares) | 310 | 7.2270 | 4.6932 | 8.7287 |
|  | *Dis* | Natural disaster rate (%) | 310 | 0.0357 | 0.0010 | 0.1414 |
|  | *Lnedu* | Weighted average of years of education (years) | 310 | 2.0314 | 1.3388 | 2.2820 |
|  | *Lnfis* | Expenditure on agriculture, forestry and water affairs (CNY 100 million ) | 310 | 6.1366 | 4.5194 | 7.1999 |

To better show the relationship of agricultural output and digital financial inclusion, we compared and ranked them at the same time. According to Table 2, we can see that the growth rate of agricultural output and digital financial inclusion rank similarly in individual regions.

**Table 2.** Comparisons and rankings of agricultural output and digital financial inclusion between regions.

| The Level of Agricultural Output | | Digital Financial Inclusion | |
| --- | --- | --- | --- |
| Six Provinces with the Highest Growth Rate | Six Provinces with the Lowest Growth Rate | Six Provinces with the Highest Growth Rate | Six Provinces with the Lowest Growth Rate |
| Tibet (11.35%) | Hebei (3.50%) | Tibet (105.95%) | Jiangsu (43.99%) |
| Guizhou (8.51%) | Liaoning (2.92%) | Guizhou (96.49%) | Tianjin (43.52%) |
| Qinghai (7.71%) | Jiangsu (2.33%) | Qinghai (95.90%) | Guangdong (40.03%) |
| Yunnan (7.15%) | Zhejiang (1.85%) | Gansu (94.89%) | Shanghai (38.41%) |
| Xinjiang (7.06%) | Shanghai (−7.00%) | Xinjiang (90.24%) | Zhejiang (38.16%) |
| Gansu (6.29%) | Beijing (−7.71%) | Yunnan (79.26%) | Beijing (37.96%) |

## 4. Results and Discussion

### 4.1. Baseline Regression Analysis

In this paper, the fixed-effects model is used to regress model (1) to test the effect of digital inclusive finance on the level of agricultural output. To reflect the independence among explanatory variables and eliminate the influence of multicollinearity, this paper gradually adds control variables in the process of regression. The baseline regression results are shown in Table 3, which show that digital inclusive finance can significantly promote the improvement of agricultural output level, and with the increase in the number of control variables, the coefficient index gradually decreases. Column (6) is the result of the regression with the inclusion of all control variables. For each unit of increase in digital inclusive finance, the level of agricultural output increases by 0.1152 units, indicating that there is a significant contribution of digital inclusive finance to the level of agricultural output, and hypothesis 1 is verified. The rapid development of technology has promoted the digitization of financial services, alleviated the long-standing financial exclusion in rural areas, reduced the financial constraints faced by farmers, increased the possibility for vulnerable groups to enjoy financial services, and provided assistance to farmers in expanding agricultural production and increasing output levels.

In terms of control variables, the crop-sown area, irrigated area, and natural disaster rate have all passed a significant level test of 5%, indicating that these control variables can significantly improve the level of agricultural output. The crop-sown area reflects the actual area of crops planted. The larger the sown area, the higher the crop yield. The appropriate amount of irrigation helps crop production and increases crop yield and the occurrence of natural disasters will affect the growth of crops and reduces crop yield. The effects of fertilizer, human capital, and financial support on the level of agricultural output are not significant. The main reason for this is that the impact of fertilizer inputs on food production has entered the stage of diminishing marginal returns, and the effect is no longer obvious [51]. In the process of urbanization, a large number of rural laborers with higher education are constantly moving to the cities, resulting in the low quality of the remaining rural laborers (according to descriptive statistics, the average time of education in rural areas is only 2 years); therefore, the role of human capital on agricultural output is limited. China is a vast country with large topographic disparities, and with the development of the rural economy and the adjustment of the rural industrial structure, the positive effect of financial support on agricultural production has been weakened [52].

**Table 3.** Baseline regression of digital inclusive finance on agricultural output levels.

| Variables | (1) | (2) | (3) | (4) | (5) | (6) |
|---|---|---|---|---|---|---|
| *Lndf* | 0.1395 *** | 0.1205 *** | 0.1284 *** | 0.1193 *** | 0.1188 *** | 0.1152 ** |
| | (0.0376) | (0.0385) | (0.0446) | (0.0416) | (0.0417) | (0.0478) |
| *Lnsarea* | 0.5147 *** | 0.4183 *** | 0.3363 ** | 0.3451 ** | 0.3453 ** | 0.3478 ** |
| | (0.1163) | (0.1221) | (0.1351) | (0.1346) | (0.1351) | (0.1370) |
| *Lnfer* | | 0.1863 | 0.1528 | 0.1645 | 0.1666 | 0.1653 |
| | | (0.1349) | (0.1234) | (0.1185) | (0.1127) | (0.1160) |
| *Lnirri* | | | 0.2188 ** | 0.2175 ** | 0.2164 ** | 0.2177 ** |
| | | | (0.1033) | (0.1006) | (0.0998) | (0.0997) |
| *Dis* | | | | −0.3862 ** | −0.3843 ** | −0.3831 ** |
| | | | | (0.1552) | (0.1580) | (0.1638) |
| *Lnedu* | | | | | 0.0423 | 0.0297 |
| | | | | | (0.2240) | (0.1926) |
| *Lnfis* | | | | | | 0.0197 |
| | | | | | | (0.0838) |
| Constant | 2.6700 *** | 2.6408 *** | 1.8612 *** | 1.7942 *** | 1.7067 ** | 1.6110 ** |
| | (0.9234) | (0.8608) | (0.5776) | (0.5785) | (0.6522) | (0.7384) |
| Province FE | Yes | Yes | Yes | Yes | Yes | Yes |
| Year FE | Yes | Yes | Yes | Yes | Yes | Yes |
| Observation | 310 | 310 | 310 | 310 | 310 | 310 |
| Adj-R$^2$ | 0.8380 | 0.8448 | 0.8527 | 0.8564 | 0.8560 | 0.8558 |
| F | 90.86 | 85.61 | 85.51 | 118.75 | 111.43 | 108.56 |

Note: ***, ** indicate statistical significant at 1%, 5%, respectively; Robust standard error is shown in parentheses.

*4.2. Robustness and Endogenous Test*

4.2.1. Robustness Test

To ensure the reliability of the baseline regression results, this paper adopts three methods to conduct robustness tests: replacing variables, excluding some provinces and cities, and controlling the fixed effect of the interaction of province and year. First, we start by replacing the dependent variables. Considering that agriculture, forestry, animal husbandry, and fishery contain intermediate inputs, this paper uses the gross output value of the primary industry instead of the gross output value of agriculture, forestry, animal husbandry, and fishery to conduct the robustness test. The results are shown in column (1) of Table 4 and the regression coefficient of digital inclusive finance is significantly positive. Second, excluding some provinces and cities. Referring to the study of Chen [46], this paper excludes provinces and cities with relatively small planting areas according to the ranking of crop cultivation areas from 2011 to 2020. The results are shown in column (2) of Table 4, where digital inclusive finance still significantly contributes to the improvement of agricultural output level. Finally, controlling the fixed effect of the interaction of region and year. Considering that the two-way fixed model of year and province is a common practice in the regression model, it may be more "flexible" and the endogenous control is not strict enough. For this purpose, control the fixed effect of the interaction of region and year. The results are shown in column (3) of Table 4, with the regression coefficients significant at the 1% level. The results of the above robustness tests indicate that digital inclusive finance can significantly increase the level of agricultural output, and also reconfirm hypothesis 1.

**Table 4.** Robustness test.

| Variables | Substitution of Dependent Variable | Eliminating Some Provinces and Cities | Interactive Fixed Effect |
|---|---|---|---|
| | **(1)** | **(2)** | **(3)** |
| *Lndf* | 0.1160 ** | 0.1174 *** | 0.0395 *** |
| | (0.0547) | (0.0252) | (0.0107) |
| Constant | 2.4197 *** | 5.0213 *** | 6.5463 *** |
| | (0.4723) | (1.0235) | (0.2974) |
| Control variables | Yes | Yes | Yes |
| Province FE | Yes | Yes | No |
| Year FE | Yes | Yes | No |
| Province-Year TE | | | Yes |
| Observation | 310 | 210 | 310 |
| Adj-R$^2$ | 0.8403 | 0.9555 | |
| F | 132.27 | 283.15 | |

Note: ***, ** indicate statistically significant at 1%, 5%, respectively; robust standard error is shown in parentheses.

4.2.2. Endogenous Test

The improvement of agricultural output level will increase farmers' operating income. On the one hand, the increase in income will mobilize farmers' enthusiasm to expand cultivation; on the other hand, it will expand farmers' spending on digital network costs and encourage farmers to further seek digital inclusive finance services so that the level of agricultural output will have an impact on the development of digital inclusive finance. The impact will lead to endogenous problems, causing parameter estimation to lose reference value. Firstly, the Durbin–Wu–Hausmann (DWH) test of the model is carried out. The *p*-value is equal to 0.0000, which rejects the original assumption that "all variables are exogenous", indicating that there is an endogenous problem. Therefore, referring to Fu [53] and Liang [54], the average distance from each province to Hangzhou (*Hang*) and internet penetration (*Net*) are selected as instrumental variables of digital financial inclusion. Further, the instrumental variables are tested to see if they satisfied the conditions of weak correlation and exogeneity. The F-statistic of the weak instrumental variable test is 79.6877, which is greater than 10; the *p*-value for the exogeneity test is greater than 0.1, indicating that the instrumental variables are reasonable. Column (1) of Table 5 reports the regression results for the first stage of the 2SLS instrumental variables. The farther away from Hangzhou, the lower the level of development of digital inclusive finance; the higher the Internet penetration rate is, the more conducive it is to promote the development of digital inclusive finance. Column (2) reports the results of the second stage of the 2SLS regression. The results show that digital inclusive finance still significantly improves the level of agricultural output, and the coefficient is significantly larger than the results of the baseline regression. According to Jiang [55], instrumental variables tend to increase the impact; therefore, the result of the increased coefficient is reasonable.

*4.3. Threshold Effect Regression*

4.3.1. Threshold Effect Test and Threshold Value Determination

To test the non-linear relationship between the development of digital inclusive finance and the level of agricultural output in the model (2), we determined the number of thresholds. In this paper, the index of digital inclusive financial development is taken as the threshold variable and the threshold effect test is conducted according to Hansen [56]. As shown in Table 6, according to the F-statistic values and the *p*-values obtained by the bootstrap, it can be seen that there is a double-threshold effect. The results of the threshold estimation are shown in Table 7. The first threshold value is 4.3746, corresponding to a digital inclusive finance index of 79.440; the second threshold value is 5.3112, corresponding to a digital inclusive finance index of 202.553. To show the authenticity and reliability of the results, the paper plots the likelihood ratio function of the digital inclusive finance threshold estimates, as shown in Figure 1. From the graph, it can be seen that the LR statistic

tends to be zero when the threshold value is within the confidence interval, indicating that the results are true and reliable.

**Table 5.** Endogenous test: 2SLS regression.

| | (1) | (2) |
|---|---|---|
| | **First Stage** | **Second Stage** |
| **Variables** | **lndigit_aggr** | *Lnoutp* |
| *Lndf* | | 1.2063 *** |
| | | (0.1887) |
| Hang | −0.0755 *** | |
| | (0.0106) | |
| Net | 0.0089 *** | |
| | (0.0010) | |
| Control variables | Yes | Yes |
| Province FE | Yes | Yes |
| Year FE | Yes | Yes |
| $R^2$ | 0.9731 | 0.9372 |

Note: *** indicate statistically significant at 1%, respectively; robust standard error is shown in parentheses.

**Table 6.** Threshold effect test.

| Threshold Number | F-Statistic Value | *p*-Value | Critical Value | | |
|---|---|---|---|---|---|
| | | | **1%** | **5%** | **10%** |
| Single threshold | 30.31 *** | 0.0000 | 20.0024 | 14.7606 | 12.5528 |
| Double threshold | 27.79 *** | 0.0067 | 21.3475 | 15.5871 | 12.3048 |
| Triple threshold | 19.82 | 0.6067 | 66.1086 | 50.5416 | 42.8227 |

Note: ***, respectively, mean significance at 1% significance levels; both the statistical value and the critical value are obtained by sampling 300 times with BS.

**Table 7.** Threshold estimation results.

| | Threshold Value | Corresponding Value | 95% Confidence Interval |
|---|---|---|---|
| First threshold | 4.3746 | 79.440 | [4.3489, 4.3844] |
| Second threshold | 5.3112 | 202.553 | [5.3059, 5.3169] |

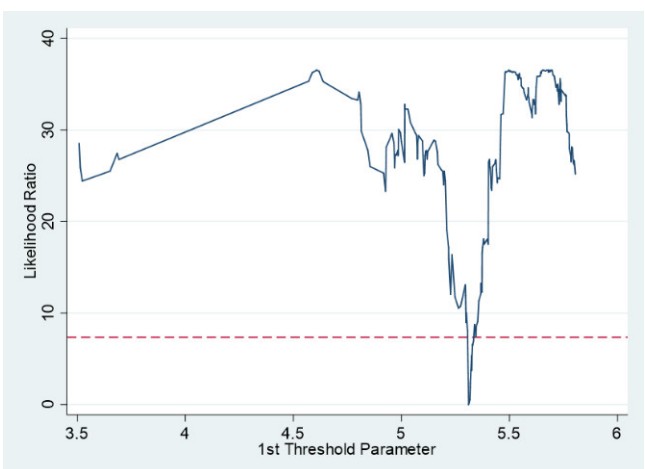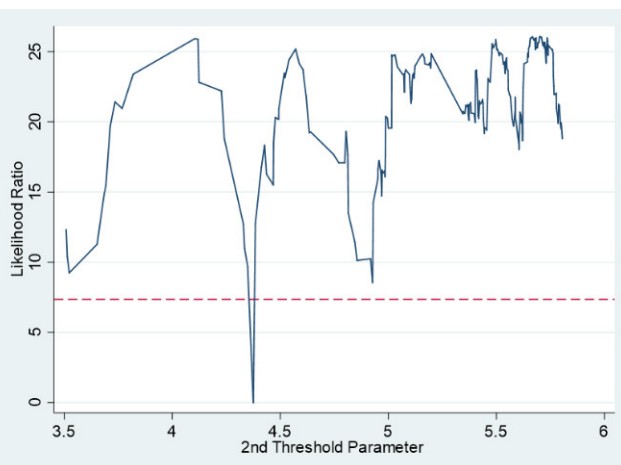

**Figure 1.** Threshold estimate and 95% confidence interval.

### 4.3.2. Analysis of Threshold Estimation Results

Table 8 reports the results of the threshold regression. When the digital inclusive financial index is below the first threshold, it can increase the level of agricultural output. Digital inclusive finance can contribute to the level of agricultural output slowly; when the index of digital inclusive financial index is between 4.3746 (corresponding value: 79.440) and 5.3112 (corresponding value: 202.553), the regression coefficient of digital finance on the level of agricultural output is 0.1376, which increases the effect. It indicates that as the level of development of digital finance increases, its marginal benefit on the level of agricultural output will increase; when the level of digital inclusive finance crosses the second threshold value, the contribution to the level of agricultural output deepens further. The impact of digital inclusive finance on the level of agricultural output has the non-linear characteristic of increasing marginal benefits. The higher level of digital inclusive finance, the greater the promotion of the level of agricultural output, which further confirms hypothesis 1.

**Table 8.** Threshold effect estimation results.

| | (1) |
|---|---|
| **Variables** | *Lnoutp* |
| *Lndf* (*Lndf* $\leq$ 4.3746) | 0.1178 *** |
| | (0.0376) |
| *Lndf* (4.3746 < *Lndf* $\leq$ 5.3112) | 0.1376 *** |
| | (0.0429) |
| *Llndf* (*Lndf* > 5.3112) | 0.1547 *** |
| | (0.0433) |
| Constant | 1.9685 *** |
| | (0.6751) |
| Control variables | Yes |
| Province FE | Yes |
| Year FE | Yes |
| Observation | 310 |
| Adj-$R^2$ | 0.8751 |
| F | 75.48 |

Note: *** indicate statistically significant at 1%, respectively; robust standard error is shown in parentheses.

### 4.4. Heterogeneity Analysis

4.4.1. Structural Heterogeneity

The breadth of coverage and the depth of usage provide the basis for analyzing the structural heterogeneity of the impact of digital inclusion on the level of agricultural output. The results of the effects of the breadth of coverage and depth of usage on output levels in agriculture are reported in columns (1) and (2) of Table 9. Both breadth of coverage and depth of usage significantly increase the level of output in agriculture, with impact coefficients of 0.0475 and 0.0835, respectively. In terms of breadth of coverage, the increase in internet penetration in rural areas helps increase the likelihood of farmers using WeChat and Alipay, increases the probability that third-party accounts are tied to bank cards, broadens the coverage of digital inclusive finance, allows more rural "long-tail" customers to be absorbed into the financial system, and provides financial support for agricultural production and operation. As far as the depth of usage is concerned, with the development of digital inclusive finance, there are more and more types of financial products and financial services driving the depth of usage of digital inclusive finance deeper and providing more options for farmers to access funds. According to the regression results, depth of usage is more helpful than breadth of coverage in improving agricultural output, mainly because of the rapid development of depth of usage. In the early stage of development, WeChat Pay and Alipay launched a series of promotional activities to attract customers to bind their bank cards. However, many users did not activate their bank cards after binding them for security reasons. As the advantages of mobile payments, such as speed, convenience, and security emerged, many accounts have been activated, which has contributed to the rapid

development of the depth of digital financial usage and increased the impact of that depth of usage.

**Table 9.** Heterogeneity test.

| | Structural Heterogeneity | | Regional Heterogeneity | | Scale Heterogeneity | |
|---|---|---|---|---|---|---|
| | Breadth of Coverage | Depth of Usage | East | Midwest | Agricultural Province | Industrial Province |
| Variables | (1) | (2) | (3) | (4) | (5) | (6) |
| Lndf | | | 0.1042 (0.0740) | 0.1048 *** (0.0349) | 0.0759 (0.0438) | 0.1231 *** (0.0261) |
| Lndf_bre | 0.0475 ** (0.0176) | | | | | |
| Lndf_dep | | 0.0835 *** (0.0264) | | | | |
| Constant | 1.7937 ** (0.7791) | 1.3885 * (0.6957) | 4.3039 *** (1.0082) | 5.1346 *** (0.8719) | 1.3491 (0.8442) | 6.4902 *** (0.9886) |
| Control Variables | Yes | Yes | Yes | Yes | Yes | Yes |
| Province FE | Yes | Yes | Yes | Yes | Yes | Yes |
| Year FE | Yes | Yes | Yes | Yes | Yes | Yes |
| Observation | 310 | 310 | 110 | 200 | 170 | 140 |
| Adj-$R^2$ | 0.8563 | 0.8551 | 0.739 | 0.964 | 0.845 | 0.976 |
| F | 212.79 | 83.57 | . | 524.65 | . | 1397.03 |

Note: ***, **, * indicate statistically significant at 1%, 5%, 10%, respectively; robust standard error is shown in parentheses.

### 4.4.2. Analysis of Regional Heterogeneity

The imbalance of regional economic development leads to regional differences in the level of development of digital inclusive finance, which, in turn, has different effects on the level of agricultural output in each region. China's developed economies are mainly concentrated in the eastern region, while the economy of the midwest is relatively backward. Due to this, this paper divides the 31 provinces into two parts, the East and the Midwest, to analyze the regional heterogeneous effects of digital inclusive finance on the level of agricultural output. The results are shown in columns (3) and (4) of Table 9: digital inclusive finance in the Midwest can significantly contribute to the increase in the level of agricultural output, while the regression coefficient in the East is not significant. The main reason may be that the economic development in the East is relatively developed, and the developed urban economy forms capital feedback to rural areas, which forms a boost to agricultural production. The rapid development of technology makes the gap in the digital inclusive finance between regions narrow gradually. Digital inclusive finance can overcome the time and space limitations of financial services, alleviate the shortage of funds in agricultural production in the Midwest, and improve agricultural output levels.

### 4.4.3. Analysis of Agricultural Scale Heterogeneity

Due to political, economic, and geographical factors, some provinces in China have developed into large agricultural provinces and some provinces have become large industrial provinces. To verify the impact of digital inclusive finance on the level of agricultural output under different agricultural production scales, this paper refers to Zhang [57]; the 31 provinces studied are divided into large agricultural and non-agricultural provinces based on the size of the share of agricultural, forestry, animal husbandry and fishery output in the national economy. The results are shown in columns (5) and (6) of Table 9: the contribution of digital inclusive finance to the level of agricultural output in agricultural provinces is significantly higher than that in industrial provinces. The reason may be that large agricultural provinces are more likely to form large-scale farming. Expanding large-scale farming will increase farmers' demand for capital, and encourage farmers to

seek bank loans. Digital inclusive finance is more likely to be accepted by large growers because it can alleviate financing constraints, lower borrowing costs, and improve service efficiency. In addition, digital inclusive finance can also provide farmers with planting technology, agricultural knowledge, policy information, and other benefits required in the process of large-scale planting.

### 4.5. Mechanism Analysis

The results of the previous baseline regressions show that digital inclusion finance can significantly increase the level of agricultural output. Referring to the mechanism analysis method from Wu [58], this section will discuss the mechanisms underlying the impact of digital inclusion finance on agricultural output levels from two perspectives: agricultural mechanization and farmers' willingness to participate in insurance.

The development of digital inclusive finance can have an impact on the level of mechanization in agricultural production. On the one hand, digital inclusive finance has the effect of increasing income. By increasing the income of farmers, it increases the use of farm equipment in agricultural production; on the other hand, digital inclusive finance has a financing effect. The development of digital inclusive finance alleviates the negative attitude of financial institutions towards supporting agriculture due to farmers' lack of collateral and information asymmetry, and increases the willingness to lend so that farmers have more funds to purchase farm equipment and promote mechanization of production. Referring to the relevant literature [59], this paper uses the integrated mechanization rate of crop cultivation, planting, and harvesting to measure the level of agricultural mechanization and regresses it on digital financial inclusion as the dependent variable. Column (1) of Table 10 reports the regression results that digital inclusive finance can significantly promote the process of agricultural mechanization. A large number of studies have confirmed the positive relationship between agricultural mechanization and agricultural output [60]. On the one hand, agricultural mechanization alleviates the labor shortage in rural areas, reduces the dependence on labor in production, and saves operation time and cost; Huang et al. [61] pointed out that agricultural machinery can replace the labor force to promote food production. When the rural labor force is transferred in large quantities, the substitution effect of agricultural machinery on the labor force is significantly improved. Peng and Zhang [62] used panel data of micro farmers to conduct their research and found that agricultural mechanization is conducive to improving the grain production efficiency of farmers. On the other hand, compared with traditional harvesting methods, agricultural machinery, and equipment can reduce losses. According to the research of Mishra et al. [63], the use of agricultural machinery harvesting can reduce the loss by 6.49%.

**Table 10.** Mechanism analysis.

| | Mechanization | Willingness to Participate in Insurance |
|---|---|---|
| **Variables** | **(1)** | **(2)** |
| *Lndf* | 0.3261 * | 0.5847 * |
| | (0.1640) | (0.3435) |
| Constant | −3.4106 | −2.7960 |
| | (2.8958) | (4.1922) |
| Control variables | Yes | Yes |
| Province FE | Yes | Yes |
| Year FE | Yes | Yes |
| Observation | 310 | 310 |
| Adj-$R^2$ | 0.4516 | 0.7758 |
| F | 11.20 | 21.30 |

Note: * indicates statistically significant at 10%; robust standard error is shown in parentheses.

The development of digital inclusive finance will have an impact on farmers' willingness to participate in insurance. Digital inclusive finance, based on the internet and big

data, can improve farmers' financial literacy and increase their knowledge of insurance, who can also break through time and geographical restrictions by participating online, increasing the convenience and accessibility of insurance participation. The results of column (2) of Table 10 show that digital inclusive finance increases farmers' willingness to participate in insurance, using the premium income of agricultural insurance as the dependent variable. There is also a large portion of literature on the impact of agricultural insurance on agricultural output [64–67]. Agricultural insurance can spread risk, optimize the structure of agricultural production, improve the efficiency of agricultural production, and promote an increase in agricultural output levels [67]. Insured farmers are more likely to invest heavily in agricultural technology and increase productivity [66]. Therefore, digital inclusive finance promotes the increase in agricultural output level by increasing farmers' willingness to participate in insurance, and H2 is verified.

## 5. Conclusions and Suggestions

Farmers often encounter both difficulties in financing and expensive financing in the production process. Digital inclusive finance relies on the internet and other technologies to provide ideas to solve the financial problem in the agricultural production process. Based on the panel data of 31 provinces and cities in China from 2011 to 2020, this paper uses a double-fixed-effect model and a threshold model to carry out a regression analysis of the impact of digital inclusive finance on the level of agricultural output. It is found that: (1) the development of digital inclusive finance can significantly increase the level of agricultural output, and the promotion effect remains significant in robustness and endogeneity tests. In addition, the effect shows a state of increasing marginal utility; (2) There is structural, regional, and production scale heterogeneity in the impact of digital inclusive finance on the level of agricultural output. Structurally, the breadth of coverage and the depth of usage can significantly improve the level of agricultural output, among which the depth of usage can be more significant; regionally, digital inclusive finance can significantly improve the level of agricultural output in the midwest regions; in terms of scale, the promotion effect of digital inclusive finance on the level of agricultural output in large agricultural provinces is more significant. (3) Digital inclusive finance improves the level of agricultural output mainly through two paths: promoting agricultural mechanization and enhancing farmers' willingness to participate in insurance.

The research of this paper confirms that digital inclusive finance can significantly promote the level of agricultural output. This conclusion is enlightening for the current development policy of digital inclusive finance. First, the government should continue to strengthen the construction of rural digital infrastructure, expand the coverage of rural networks, and promote the development of digital inclusive finance in rural areas. At the same time, the government should also pay attention to the training of farmers' digital skills to improve the digital literacy and application ability of farmers. Second, governments should implement the regionally differentiated development strategy of digital inclusive finance. In particular, the midwest regions and major agricultural provinces should formulate corresponding policies to deepen the digital construction of traditional financial institutions, help rural development, and open up the "last kilometer" of agricultural production. Third, financial institutions, especially township banks, should accelerate the digitization of traditional financial services, innovate the types of financial products, and meet the diversified needs of farmers by increasing the supply categories of financial products. For example, by providing special agricultural loan products, increasing subsidies for agricultural machinery and equipment, etc. Insurance institutions should also: improve the agricultural insurance system, actively expand the coverage of rural insurance, improve the compensable capacity of agricultural insurance, consolidate and deepen the role of agricultural insurance as a "stabilizer" in agricultural production, and guide farmers to participate in agricultural insurance. Fourth, accelerate the process of agricultural mechanization and promote the modernization of agricultural production. It is necessary to increase investment in agricultural scientific research, promote the application

and popularization of agricultural production technology, improve the level of agricultural technology, and promote the efficient development of agricultural production.

There are also some limitations in this study. On the one hand, the sample size is relatively small. Considering the availability of some data, the paper selects the provincial panel data from 2011–2020 for analysis, which is easily affected by uncertain factors, such as emergencies in a certain year, and affects the accuracy of empirical results. In the future, the city-level panel data can be used to increase the research sample to make up for the possible impact of the relatively small number of samples in the study. On the other hand, digital inclusive finance has obvious spatial agglomeration in the development process. Whether it will have a spatial spillover effect on agricultural output level can be further discussed in future research.

**Author Contributions:** J.W., conceptualization, methodology, formal analysis, resources, data curation, writing; S.X., verification, review and supervision. All authors have read and agreed to the published version of the manuscript.

**Funding:** This work was financially supported by the National Social Science Foundation major project (18VHQ003).

**Institutional Review Board Statement:** Not applicable.

**Informed Consent Statement:** Not applicable.

**Data Availability Statement:** Not applicable.

**Conflicts of Interest:** All authors declare no conflict of interest.

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
