# Peer review of "The Impact of Digital Financial Inclusion on the Level of Agricultural Output"

_sustainability, doi:10.3390/su15054138_

Round 1
Reviewer 1 Report
The logical structure of the paper is good, but I see a problem in the fact that the Introduction part is not delimited by the Literature Review.
Therefore, the introduction is too long.
I suggest the delimitation of the two sections.
Author Response
Dear reviewer,
Thank you for your generous comments and valuable suggestions in reviewing my thesis during your busy schedule. These comments have been very helpful. We have read these comments carefully and have revised the paper one by one in accordance with the revisions and feel that the quality of the paper has improved considerably as a result of your revisions, the modifications are shown below. Revisions to the manuscript have been marked up using the “Track Changes” function. We have uploaded the revised manuscript file, you can use the revision mode directly to see where the paper has been revised and would like to receive your comments.
Point 1: The introduction is too long, and the Introduction part is not delimited by the Literature Review.
Response 1: Thanks for pointing out the mistake. We agree with the reviewer that the introduction is too long. The part of introduction including literature review, which leads to the introduction being too long, and the literature review is not concise and comprehensive enough. Thus, we revised the citation of the manuscript. On the one hand, we sorted out previous relevant researches, and clarified the research motivation. On the other hand, the literatures we cited are the latest ones; We introduced the innovation and contribution of this study in advance, so that readers can understand the research theme of this paper clearly; Finally, the introduction of the following framework was supplemented.
Thanks again for your time and consideration. We really appreciate your efforts in reviewing our manuscript during this unprecedented and challenging time. We wish good health to you and everything goes well with your work!

Reviewer 2 Report
Dear Author(s),
thanks to having submitted your paper titled "The Impact of Digital Financial Inclusion on the Level of Agricultural Output".
I think your paper has good potential, but some elements should be improved.
For instance, your abstract lacks fluency. My suggestion is to rewrite your abstract, enhancing the fluency and making it less schematic and more discursive. Also, include from here what your theoretical background (TB) is, to let the reader know where you are advancing knowledge.
Then, your introduction is very long. Thus, try to shorter this section, stressing also here your TB.
My bigger concern refers to theoretical analysis. This section should be improved, starting by creating an ad hoc section on your TB. Then divide each paragraph+ hypothesis into subsections, where you state critically the literature review you take from to formulate your hypotheses. Here citations of recent references are crucial! Thus I highly recommend inserting citations and recent sources to update these sections.
Endly, I highly suggest creating three ad hoc sections after your conclusion to emphasize (1) theoretical and managerial implications, (2) limits of this study, and (3) future directions.
Good luck with this review!
I thank the Editor to have chosen me as Reviewer.
Author Response
Dear reviewer,
Thank you for your generous comments and valuable suggestions in reviewing my thesis during your busy schedule. These comments have been very helpful. We have read these comments carefully and have revised the paper one by one in accordance with the revisions and feel that the quality of the paper has improved considerably as a result of your revisions, the modifications are shown in attachment. Please see the attachment.

Reviewer 3 Report
Dear author,
I have read this paper with interest, and I hope you are open to some comments that might clarify or even improve this presentation a bit. You showed very interesting research in the paper. But in my opinion, a few things are missing.
The authors try to prove the significant impact of DFI on the level of agricultural production. However, I'm not sure that this has been achieved as the studies described claim. Admittedly, there is an attractive research method with statistically significant results, but there is no more information. In my opinion, there is a lack of a broad description of the research area and specific examples, as well as simple descriptive statistics (e.g., coefficient of variation). There is very important information for the international reader. In my opinion, some of the variables are also inappropriately selected. As a result, they should not be used for comparison. These are absolute variables (e.g., Lnoutp, Lnsarea, Lnirri) for a given region. This results in, as the authors themselves write: "The larger the sown area, the higher the crop yield". In other words, the larger the agricultural production area of the region, the higher the crop yields, or the higher the agricultural expenditures and production value. But what does this look like on a per-hectare or per-farm basis, which allows for real comparisons? A better solution seems to be to use variables with relative values, such as per hectare or per farm. It may be that in a relatively small region, there are few farms, but they are large, mechanized and developed. However, they produce a smaller total value of production (or not). A larger region may have more of everything, including farms, but they are economically weak. This is actually not mentioned in the article. Another thing is that quite attractive, econometric methods of analysis indicate a relationship between variables. But can we really speak of a cause-and-effect relationship? It may be a statistically significant correlation, but without comparative studies with other variables, it is difficult to confirm. It may be that the phenomena occur side by side, but there is no relationship between them. Or the impact is weak if the analysis takes into account the level of fertilization, soil quality, plant varieties, the amount of pests, diseases or weeds, etc. It is a pity that such a wide range of issues were not included.
Abstract
The abstract introduces the manuscript well and concisely. But
Very important conclusion "the level of digital inclusive finance, the higher the level of agricultural production". But it should be very carefully explained in the main text what it is.
Keywords: Add "China"
Introduction:
Keep in mind - The introduction should briefly place the study in a broad context and highlight why it is important. It should define the purpose of the work and its significance, including specific hypotheses being tested. The current state of the research field should be reviewed carefully and key publications cited. Please highlight controversial and diverging hypotheses when necessary. Finally, briefly mention the main aim of the work and highlight the main conclusions. Keep the introduction comprehensible to scientists working outside the topic of the paper.
So merge the two chapters - the Introduction and Chapter 2. This will eliminate repetitive content.
The first two chapters contain many paragraphs without sources or citations, such as "Central Document No. 1" or "which reached a record high, indicating that China's dependence on foreign food is also increasing. In addition, the emigration of young workers, the aging of the population, the reduction of arable land, and the control of seed sources by foreign investors are hindering the development of agriculture. It is of great practical significance to increase food production and stabilize the supply of agricultural products in order to consolidate the achievements of poverty alleviation and realize the strategy of rural revitalization".
For footnotes 1, 2, and 3, please cite specific sources of information with titles and possibly authors and when these publications were used. Footnotes 1, 2 and 3 should be converted to a bibliography reference.
H2a and H2b - why were only these factors analyzed? Doesn't the easier availability of funds make it possible to increase the use of fertilizers, crop protection products or more resistant plant varieties?
Materials and Methods
They should be described with sufficient detail to allow others to replicate and build on published results. New methods and protocols should be described in detail while well-established methods can be briefly described and appropriately cited. Give the name and version of any software used and make clear whether computer code used is available. Include any pre-registration codes.
The first subsection should be a description of the study area - 31 regions and their characteristics in terms of agriculture.
The second subsection should be a detailed description of the variables and their descriptive statistics. For example, what is the "Digital Inclusive Finance Index (2011-2020) measured by the Digital Finance Research Center of Peking University" (specifically)? How was this index adapted to the study?
The third chapter is a description of the analysis methods, software and weaknesses of these methods.
Present local monetary values currency in an international currency such as $. This will be more understandable to the international reader. For example, 7 833.951 billion Yuan = $?
“some variables are logarithmically processed” – which?
I wish the authors had done comparisons and rankings between regions. It would be possible to observe the direction and speed of change in individual regions.
Conclusions and Suggestions
"Digital inclusive finance improves the level of agricultural production mainly through two pathways: promoting agricultural mechanization and increasing farmers' willingness to participate in insurance." Well, it's hard to come to a different conclusion if only those two were studied.
Reread the manuscript and check for grammatical and punctuation errors.
So, after summarizing the comments and suggestions contained in this review and after approval by the editor, the manuscript can be published.
Hope this helps, keep up the good work!
Author Response

(The authors gave the same response as above.)

Reviewer 4 Report
This paper examines the impact of digital finance on agricultural output. Results indicate that digital financial inclusion could significantly improve the level of agricultural output. Although this contains some merits and has incremental contributions to the literature, the present study has major concerns.
My comments are as follows.
1. The paper, in its current form, lacks clear motivation. I would suggest further work on highlighting the research gap and better motivating the research question. I suggest authors clarify the differences between their research and previous evidence.
Refs:
1. Hong, M., Tian, M., & Wang, J. (2022). Digital Inclusive Finance, agricultural industrial structure optimization and agricultural green total factor productivity. Sustainability, 14(18), 11450.
2. Hu, Y., Liu, C., & Peng, J. (2021). Financial inclusion and agricultural total factor productivity growth in China. Economic Modelling, 96, 68-82.
3. Ji, X., Wang, K., Xu, H., & Li, M. (2021). Has digital financial inclusion narrowed the urban-rural income gap: the role of entrepreneurship in China. Sustainability, 13(15), 8292.
4. Liu, Y., Luan, L., Wu, W., Zhang, Z., & Hsu, Y. (2021). Can digital financial inclusion promote China's economic growth?. International Review of Financial Analysis, 78, 101889.
5. Ma, J., & Li, Z. (2021). Does Digital Financial Inclusion Affect Agricultural Eco-Efficiency? A Case Study on China. Agronomy, 11(10), 1949.
6. Wang, X., & He, G. (2020). Digital financial inclusion and farmers’ vulnerability to poverty: Evidence from rural China. Sustainability, 12(4), 1668.
7. Yue, P., Korkmaz, A. G., Yin, Z., & Zhou, H. (2022). The rise of digital finance: Financial inclusion or debt trap?. Finance Research Letters, 47, 102604.
8. Zhou, Z., Zhang, Y., & Yan, Z. (2022). Will Digital Financial Inclusion Increase Chinese Farmers’ Willingness to Adopt Agricultural Technology?. Agriculture, 12(10), 1514.
2. The introduction part is not enriched with the latest studies. Please revise it thoroughly and make it succinct. The paragraphs must need to be interconnected.
3. Although this paper has potential, it fails to present substantial evidence. The authors might want to improve the research design and methodology. The authors may want to explain better how digital finance works.
4. The authors may also need to control the fixed effect of the interaction of region and year.
5. Conclusion would benefit a lot from the theoretical discussion.
Author Response

(The authors gave the same response as above.)

Round 2
Reviewer 2 Report
Dear Author(s),
your manuscript benefits from the previous round of review, but I think you can work more on two crucial points, which are weak and not acceptable in the present form.
1) Readability: please send the manuscript to professional proofreading to improve the quality of the language.
2) Theoretical and managerial implications are absent, so please work on them to show your contribution. Then, extend the propositions you suggested (are they propositions or managerial implications), and explain for instance how and who can promote the development of digital inclusive finance in rural areas (and followings).
Good luck with this review!
Author Response
Dear reviewer,
Thank you for your generous comments and valuable suggestions on my paper during your busy schedule. These comments are very helpful. We have carefully read these comments and revised the paper one by one according to the revised content, and believe that the quality of the paper has been greatly improved due to your revision. Please see the attachment for details of the modification.

Reviewer 3 Report
Dear Authors,
The authors should better explain why absolute variables (e.g. total output value of agriculture, forestry, animal husbandry and fishery production; sown area) rather than relative variables (e.g. number of farms in the region; average output value per farm; average sown area, etc.) were included. This needs to be justified with reference to the literature.
Hope this helps, keep it up!
Author Response

(The authors gave the same response as above.)

Reviewer 4 Report
Make sure to format the paper according to the journal requirements.
Author Response

(The authors gave the same response as above.)

Round 3
Reviewer 2 Report
I am happy with the author(s) revisions and I have no more changes to suggest.